# Laurel Regeneration Management by Smallholders to Generate Agroforestry Systems in the Ecuadorian Amazon Upper Basin: Growth and Yield Models

Álvaro Cañadas-López [1,*], Paul Gamboa-Trujillo [1], Santiago Buitrón-Garrido [1], Byron Medina-Torres [1], Christian Velasco [2], José de Jesús Vargas-Hernández [3] and Christian Wehenkel [4]

1   Carrera de Ingeniería en Recursos, Naturales Renovables, Facultad de Ciencias Biológícas, Universidad Central del Ecuador, Quito 170150, Ecuador
2   FAO-PRO Amazonía, Organización de las Naciones Unidas Para la Alimentación y la Agricultura (FAO), Quito 170518, Ecuador
3   Posgrado en Ciencias Forestales, Colegio de Posgraduados, Montecillo 56230, Mexico
4   Instituto de Silvicultura e Industria de la Madera, Universidad de Juárez del Estado de Durango, Durango 34120, Mexico
*   Correspondence: agcanadas@uce.edu.ec; Tel.: +59-39-3906-8644

**Abstract:** Laurel (*Cordia alliodora* Ruiz & Pav. Oken) is a Neotropical native tree that is easily regenerated in the secondary forest within the Amazon region. Amazonian smallholders use this tree regeneration to obtain a homogeneous forest cover when developing local agroforestry systems, which do not depend on nursery seedling production for tree planting. The objective of the present investigation was to develop growth and yield models for Laurel within the local agroforestry systems. A total of 226 sampling plots were measured between 2010–2011 and 2014–2015. Chapman- Richard, Hosslfeld II, and the generalized algebraic difference approach (GADA) form of the Chapman-Richard's function was used for modeling height-age and diameter-age relationships. Eight volume models were tested to describe total stem volume. The GADA method was suited to describe the Laurel height and diameter-age growth. The cutting cycle for agroforestry systems with a density of 300 trees ha$^{-1}$ at the best site index (SI) (22 m) produced 13.9 m$^3$ ha$^{-1}$ year$^{-1}$ and a total wood yield of 195.1 m$^3$ ha$^{-1}$ at age 14. In the worst SI (14 m), the average annual yield was 3.5 m$^3$ ha$^{-1}$, with a total yield of 83.3 m$^3$ ha$^{-1}$ at age 24 years. The Spurr potential model was the best fit to describe the volume of the Laurel according to the Akaike information criteria. The Laurel biological (optimal) rotation age suggests that the minimum cutting diameter should be lowered from 30 cm of DBH in the research zone. Management of the natural regeneration of secondary forests by smallholders is a local agroforestry practice that should be given greater attention, especially within protected forest areas.

**Keywords:** *Cordia alliodora*; DBH growth curve; forest management; site index models; total volume

## 1. Introduction

Agrosilvopastoral systems are long-established and geographically widespread strategies developed by Amazonian indigenous peoples to adapt to the dynamics of Neotropical humid rainforests [1]. These agroforestry systems have demonstrated their resilience and potential as an alternative land use adapted to the contemporary realities of sudden changes market [2]. Due to the high structural recuperation of neotropical forests after conversion to pasture and agriculture, authors such as Cañadas [3] emphasized that the economic value of secondary forests could be increased by establishing agroforestry systems and that the economic value of fallow land could be improved by managing Laurel (*Cordia alliodora* Ruiz & Pav. Oken) from natural regeneration. Laurel is a heliophilous species that occurs in primary forest gaps and regenerates in secondary forests [4]. Its wood quality meets the

requirements of an ever-growing industry. According to Velázquez [5], Laurel presents excellent qualities for pulp production, and Ecuador's imports of pulp increased from 334,660 t in 2016 to 356,000 t in 2019, experiencing a decrease to 254,000 t in 2020 [6]. The species appears suitable for agroforestry systems, especially because of its low production costs and role in biodiversity conservation [3].

Traditional indigenous agroforestry systems represent technologies that evolved along with the domestication of Amazonian Forest species and their use in agriculture, contributing to an important cultural heritage still preserved by many Amazonian tribes [7]. Thus, glimpsing tree growth is the fundamental basis for sustainable management and for predicting responses of different landscapes to various management or climate change scenarios [8]. Porro et al. [9] pointed out that agroforestry systems in the Amazon region have not been sufficiently evaluated, especially the tree component. Thus, there is a lack of technical information such as site indices, growth rates, and rotation age to support the economic viability of agroforestry systems in the Amazon. Subsequently, da Silva et al. [10] emphasized that information on the growth and production of forest components in Amazonian agroforestry systems of any species remains scarce in the region.

One such unknown forest parameter is the site index (SI) concept. This reflects the effect of environmental factors that influence the growth of trees in a given stand. The role of site quality on forest productivity can be estimated using different indices. For even-aged stands, the SI is a forest parameter that allows estimating tree growth and yields at different tree ages [8,11,12]. Nevertheless, the stand events history influences total tree height, growth, and SI [13]. Forest models provide reasonable growth and development predictions of forest stands. Several authors, such as Alder and Montenegro [14]; Ricker and Río [15]; Olschewski and Benítez [16–18]; de Koning et al. [19,20]; Parresol and Devall [21]; and Somarriba et al. [22], among others, developed the Chapman-Richard's anamorphic models for Laurel diameter and height growth in different countries and growing conditions.

The generalized algebraic difference equation (GADA) method offers multiple advantages for SI curves because the equation can be expanded according to the various growth theories, such as growth rate and asymptote. Thus, it allows two parameters to be SI-specific, the developed dynamic function to be more flexible, and polymorphic SI equations with multiple asymptotes can be obtained [23–26]. Additionally, determining individual tree volume is a relevant task in dendrometry. There are many difficulties in directly establishing tree volume by cubing sections. Hence, it is of great importance to have a mathematical equation to determine this forest parameter in an indirect way through simple measurements such as the diameter and total height of a standing tree [27,28]. With these measurements, the average annual increment and periodic annual increment can be calculated to establish the biologically optimum rotation age (BORA) for any tree species [29].

In this study, it is highlighted as a research hypothesis that the fit of the SI curve for height and the diameter at breast height (DBH)-age growth curve with the GADA method applied to the Chapman-Richard function is equal to the fit obtained by the basic Chapman-Richard and Hossfeld II models. It is also hypothesized that the fit parameters are equal for the eight models studied to estimate the growth and yield of Laurel in traditional agroforestry systems in the Upper Basin of the Ecuadorian Neotropical Amazon. The current research aims to analyze the implications of the estimated productivity of Laurel from the models developed for the management and permanence of these traditional agroforestry systems.

## 2. Materials and Methods

### 2.1. Study Area

The study was conducted in the canton Loreto, Orellana province (Figure 1), with an elevation above sea level between 500 to 1200 m. The average annual temperature in the sampled region is 21 °C. The mean annual precipitation is 4000 mm, which is evenly distributed over the year [30]. Soils belong to the large group of Hydrandepts. Their

high-water retention capacity causes severe leaching and weathering of primary minerals, releasing aluminum and hydrogen cations and contributing to soil acidification [3]. The land tenure status within the canton of Loreto is distributed as follows: 14.1% belongs to the State and is occupied by the Sumaco and Galeras National Park. Additionally, 73.2% have a communal title, but land use is individual among community members, and 12.7% is for private use. Meanwhile, 16% of Loreto canton is under a protected forest status, belongs to the State, is divided into communities, and the use is individual [31,32].

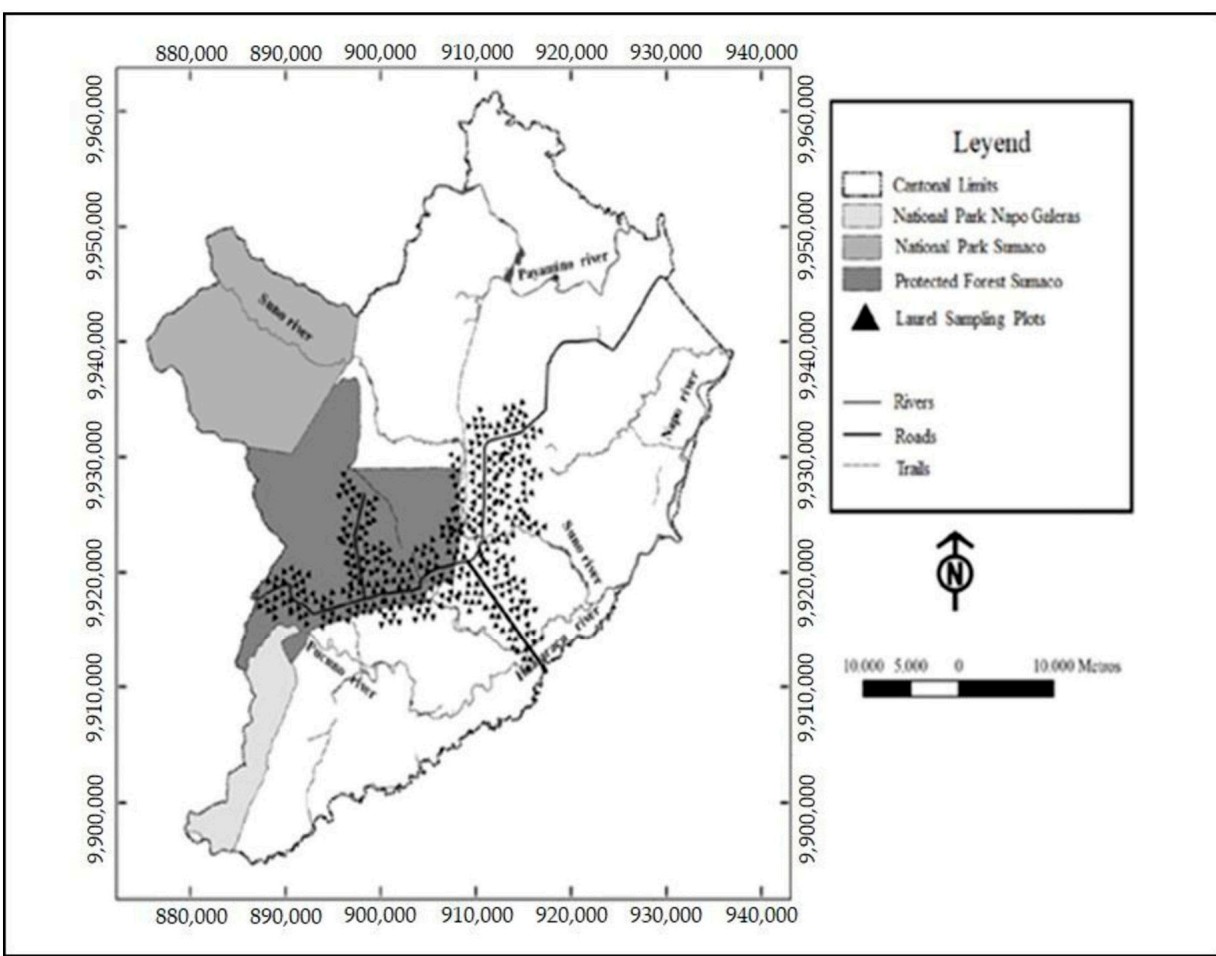

**Figure 1.** Laurel agro-forests sampling plots and spatial distribution in the research zone, Amazonian upper basin.

Neotropical forest areas are deforested to be used for cattle and agricultural purposes. Once these areas become less productive, they are abandoned for a period of 10–15 years in the Ecuadorian Amazon upper basin to give way to natural forest regeneration, which occurs in different phases of secondary growth [33,34]. Amazon smallholders use secondary forest regeneration to form traditional agroforestry systems, and it consists of cutting one hectare of primary forest with some Laurel trees being left to reinforce the natural regeneration of this species. The area was used in the cultivation of maize (one year) (*Zea mays*), naranjilla (four years) (*Solanum quitoense* Lam.), and Robusta coffee (*Coffea canephor* Pierre ex A: Froehner), with Laurel regeneration progressing between crops. Thus, a relatively homogenous forest was established, resulting in a traditional agroforestry system [30], and this is the type of agroforestry system where the research plots were established.

## 2.2. Tree Competition Index for DBH Analysis in Agroforestry Systems Amazon Upper Basin

For both the Laurel diameter growth curve and total volume modeling in agroforestry systems, a competition factor between trees should be considered. Since the information generated was based on age, tree number ha$^{-1}$, diameter, and height increments, the Hart–Becking competition index or also known as relative growth spacing index (*RS*), was used consisting of the average horizontal distance between trees expressed as a percentage of the dominant height [35] and its equation is:

$$RS = \left( \frac{\sqrt{10000/N}}{H} \right) \times 100 \tag{1}$$

where, *N* = stems per ha, *H* = dominant tree height (m). This index can be used independently of stand age and can be adapted for regular and irregular stands [36]. *RS* was related to diameter increment as a measure of competition factor between trees in the sampled agroforestry systems.

## 2.3. Site Index

A total of 226 sampling plots were assessed to obtain forestry data. The first criterion for the distribution of the research plots was along the roads, and the second criterion was the distance from the road edge to the hinterland of 2000 and 3000 m. Because the Laurel stands came from natural regeneration presenting a wide range of densities, the rectangular sample plot size varied between 3000 m$^2$ and 5000 m$^2$ (compared with [4,37,38]) to obtain a minimum of 50 Laurel trees per plot. Each rectangular plot was georeferenced, coded, and the age of the agroforestry system was obtained from the agroforestry owner. Smallholders keep track of the age of the agroforestry system because cash crops are grown under this system and because forest felling is performed in minga (joint work with families, food, and feast). Total tree height was measured with a Haga hypsometer and DBH with a caliper. Measurements were carried out between 2010–2011 and 2014–2015. The measurement interval in the present study has been properly spaced in time to absorb abnormal climatic extremes, generally five years, according to Tewari et al. [29]. The average tree density in the research area was 298 trees ha$^{-1}$ (min: 80.0; max: 398.4; SE $\pm$ 3.9). Table 1 presents the basic information of the stand plots and Laurel trees for the adjustment of the volume models in the research zone.

**Table 1.** Characterization in age, DBH, and H data in the research plots and Laurel tree data for volume fitting in the Amazonian upper basin.

| Characterization of the Data Used in the Fitting Process | Variable | Average | Maximum | Minimum | Standard Error ($\pm$) |
|---|---|---|---|---|---|
| Stand information of 226 plots | Age in years | 15.10 | 30.00 | 1.00 | 0.55 |
| | Diameter at breast height (cm) | 18.30 | 31.45 | 0.98 | 0.46 |
| | Total height (m) | 16.68 | 28.41 | 0.50 | 0.39 |
| Laurel sample tree for volume fitted (195 trees) | Diameter at Breast height (cm) | 16.97 | 30.24 | 4.30 | 2.91 |
| | Height (m) | 14.03 | 30.23 | 5.00 | 1.87 |
| | Volume (m$^3$) | 0.25 | 0.99 | 0.02 | 0.01 |

The Laurel tree's age corresponds to the age of the agroforestry system, and to determine the SI, a minimum of 50 tallest Laurel trees in each sample plot were measured and expressed as a mean dominant height (*H*) and *DBH*. The model functions were Chapman-

Richard's, Equation (2) [39], Hosslfeld II, Equation (3) [40], and the generalized algebraic difference approach (GADA) using Chapman-Richard's, Equations (4) and (5) [24].

$$H = a_1\left(1 - \exp^{(-a_2 t)}\right)^{a_3} \quad DBH = a_1\left(1 - \exp^{(-a_2 t)}\right)^{a_3} \tag{2}$$

$$H = \frac{t*\left(a+b*t^2\right)}{c+t^3}$$
$$DBH = \frac{t*\left(a+b*t^2\right)}{c+t^3} \tag{3}$$

where:

$H$ = dominant height (m)

$DBH$ = dominant diameter at breast height

$t$ = age

$a_1, a_2, a_3$ = equations parameters

The guide curve method was employed to generate a SI function that described the average pattern of the complete series of $H$ and $DBH$ growth without differentiating between zones in the study area [41–43]. Polymorphic curves were developed from this curve, and the $H$ attained at 10 years of age was established as a reference for the Laurel in the Ecuadorian Neotropical Amazon Upper Basin. To derive a polymorphic model with multiple asymptotes from Chapman Richard's model (Equation (2)) with the GADA formulation requires that the asymptote $a_1$, as well as the shape parameter, $a_3$, can be considered dependent on the SI ($X$) and this relationship can be expressed as follows:

$$a_1 = X, \ a_3 = b_2 + \frac{b_3}{X} \ and \ a_2 = b_1 \tag{4}$$

For the two pairs of $H_d$ values, Equation (2) is rewritten as:

$$H_0, t_0 \rightarrow H_0 = e^x \cdot \left(1 - exp^{-b_1 \cdot t_0}\right)^{b_2 + \frac{b_3}{X}}$$
$$H_1, t_1 \rightarrow H_1 = e^x \cdot \left(1 - exp^{-b_1 \cdot t_1}\right)^{b_2 + \frac{b_3}{X}} \tag{5}$$

where $H_0$ is the dominant heigh at the initial age $t_0$ and $H_1$ is the dominant height at age $t_1$ and equation's parameters $b_1, b_2, b_3$. By taking logarithms (ln) on both sides of the first of the above equalities, the following equality is obtained:

$$ln(H_0) = X + \left(b_2 + \frac{b_3}{X}\right) \cdot \ln\left(1 - exp^{-b_1 \cdot t_0}\right) \tag{6}$$

For Equation (5), the solution of $X$ involves finding the roots of a quadratic equation and choosing the most appropriate root expression, and then substituting it into the second expression of the pair of Equation (3). Rearranging the terms of Equation (6) yields a polynomial of second degree in the function of $X$, the value of which can be cleared. The solution for $X$ in Equation (6) with similar condition values $t_0$ and $H_0$, is:

$$X_0 = \frac{1}{2}\left\{lnH_0 - b_2 L_0 \pm \sqrt{[lnH_0 - b_2]^2 - 4b_3 L_0}\right\} \tag{7}$$

where

$L_0 = ln\left[1 - exp^{-b_1 t_0}\right]$.

By selecting the appropriate solution of $X_0$ and substituting it into the second equality of Equation (5), the following dynamic equation in algebraic differences which gives polymorphic curves with multiple asymptotes:

$$H_1 = H_o \left[ \frac{1 - exp^{-b_1 t_1}}{1 - exp^{-b_1 t_0}} \right]^{(b_2 + b_3 / X_0)} \tag{8}$$

where $X_0$ is given in Equation (7), the equation to real dominant height-age data allows us to estimate the values of the global parameters $b_1$, $b_2$, and $b_3$. All families of curves obtained with the GADA method are invariant with respect to the reference age and invariant with respect to the simulation path [23–26]. The simultaneous adjustment of the mean structure (given by the growth equation) and the mean error structure (given by the autoregression model) was performed with the GADA procedure within the R program [44] (version 2.2.2) (R Foundation for Statistical Computing, Vienna, Austria, 2021). The procedure was applied for DBH to generate the tree mean dominant DBH growth models.

The estimated models for Laurel *H* and *DBH* growth were compared using the graphical analysis of bias and root mean squared error (RMSE) parameters describing the fit and quality of a model. Graphical comparisons are necessary to ensure the curves fit the data over their full range. In addition, different models may have the same comparison statistics but different responses. Graphical analysis was carried out by (1) superimposing the fitted curves, (2) plotting residuals against values predicted by the model, and (3) analyzing changes in bias and RMSE for the different age groups [4,34].

### 2.4. Volume Estimation

For Laurel volume estimation, 195 dominant trees were chosen and destructively sampled from the 195 sites (Laurel tree statistics are summarized in Table 1). The Laurel tree selection criterion was the greatest height and best shape corresponding to each age of the agroforestry system, trying to cover all possible diameter classes. A diameter tape was used to measure the over-bark tree diameter ($d_i$) at ground level and at different heights: 0.3 m, 2.3 m, and every 2.0 m along the stem up to the top. The total tree volume for each tree ($v$, m$^3$) was calculated using the length ($l$) and diameter at each end of the section ($d_i$ and $d_{i+1}$) of the felled specimens using the following formula [27]:

$$v = \frac{l\pi}{3} \left[ \left( \frac{d_i}{2} \right)^2 + \frac{(d_i d_{i+1})}{4} + \left( \frac{d_{i+1}}{2} \right)^2 \right] \tag{9}$$

Eight models for estimating the volume were tested [4,37,38] with the purpose of establishing the best regression model between $v$, H, and DBH for Laurel agroforestry stands (Table 2). Parameters $a$, $b$, $c$, and d were generated by the generalized moment method (GMM) implemented in the Statistica program to adjust the volume models [45].

**Table 2.** Tested models for fitting volume equations for Laurel trees using diameter at breast height (DBH in cm) and total tree height (H in m) of dominant trees in the Amazonian upper basin.

| Model | Expression | Equation |
|---|---|---|
| Schumacher–Hall (allometric) [46] | $v = a \cdot DBH^b \cdot H^c$ | (10) |
| Spurr [47] | $v = a \cdot DBH^2 \cdot H$ | (11) |
| Spurr potential [47] | $v = a \cdot \left( DBH \cdot H \right)^b$ | (12) |
| Spurr with an independent term [47] | $v = a + b \cdot DBH^2 \cdot H$ | (13) |
| Incomplete generalized combined variable [48] | $v = a + b \cdot H + c \cdot DBH^2 \cdot H$ | (14) |
| Australian formula [49] | $v = a + b \cdot DBH^2 + c \cdot H + d \cdot DBH^2 \cdot H$ | (15) |
| Honer [50] | $v = DBH^2 / (a + b/H)$ | (16) |
| Newnham [51] | $v = a + b \cdot DBH^c \cdot H^d$ | (17) |

The performance of the proposed volume models was established through their mean square error (MSE), standard error (SE), and determination coefficient ($R^2_{Adj}$). The Akaike information criterion (AIC) was used to establish the best relative model. The lower the AIC value, the better the model [52]. The calculation of Laurel yield is based on the best model for stem volume. This model was used to estimate wood production within traditional agroforestry systems. The volume per hectare was obtained by multiplying the number of trees per hectare (N) and the modeled average tree volume per age and SI ($v_i$).

$$v_{ha} = Nv_i \tag{18}$$

Mean annual increment (MAI) was assessed as $v_{ha}$ at harvesting time divided by the stand age ($t$) at rotation length.

$$MAI = \frac{v_{ha}}{t} \tag{19}$$

The periodic annual increment (PAI) model determined the v change between the beginning and the end of a growth period, divided by the number of years.

$$PAI = \frac{v_{ha2} - v_{ha1}}{t_2 - t_1} \tag{20}$$

$v_1$ and $v_2$ are volume per hectare at time 1 ($v_{ha1}$), and at time 2 ($v_{ha2}$), respectively, and $t_1$ and $t_2$ correspond to years starting and ending the growth period. The Laurel BORA was established when PAI and MAI were equal, and MAI was the maximum for Laurel.

## 3. Results

### 3.1. Tree Competiton Factor in Agroforestry Systems

Figure 2 shows the non-significant relationship between *RS* (%) and diameter increment (cm). About 10% of the investigated agroforestry systems are between *RS* 20 and 30% and would show slight tree lateral competition.

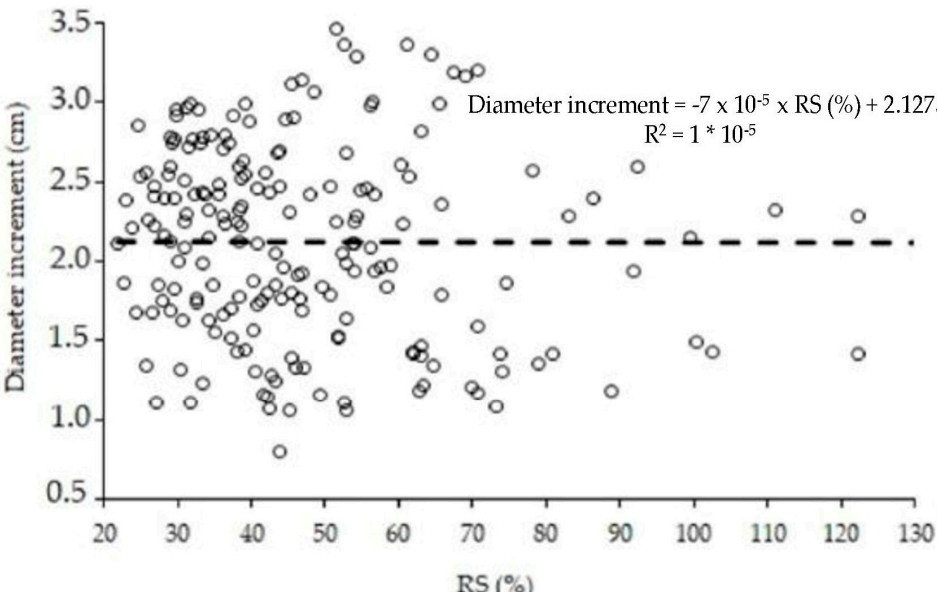

**Figure 2.** Relationship between *RS* (%) and diameter increment for 226 sampling plots in the upper Amazonia basin.

### 3.2. Growth Models for Laurel Height and Diameter-Age Growth

Table 3 resumes parameter estimation as well as adjustment statistics of models for *H*-age and *DBH*-growth relationships (Equations (2), (3) and (8)). A statistically significant

*p*-value was found for all estimated parameters of the three analyzed models; the lowest RMSE was obtained with Equation (8).

**Table 3.** Estimated values of parameters, *p* values, and goodness-of-fit statistics for the three height-age and diameter-age models for Laurel agroforestry systems in the Amazonian upper basin.

| Parameter | Model | Parameter | Estimated Value | Standard Error | *p*-Value | RMSE |
|---|---|---|---|---|---|---|
| Height-age | Equation (1) | $a_1$ | 41.748 | 16.86 | <0.001 | 3.17 |
| | | $a_2$ | 0.015 | 0.02 | <0.001 | |
| | | $a_3$ | 0.550 | 0.07 | <0.001 | |
| | Equation (2) | $a_1$ | 0.181 | 0.24 | <0.001 | 3.22 |
| | | $a_2$ | −0.133 | 0.08 | <0.001 | |
| | | $a_3$ | 0.130 | 0.12 | <0.001 | |
| | Equation (8) | $b_1$ | 0.109 | 0.03 | <0.001 | 2.92 |
| | | $b_2$ | −12.593 | 3.50 | <0.001 | |
| | | $b_3$ | 42.093 | 11.81 | <0.001 | |
| Diameter-age | Equation (1) | $a_1$ | 154.363 | 2513.90 | <0.001 | 3.11 |
| | | $a_2$ | 0.0005 | 0.02 | <0.001 | |
| | | $a_3$ | 0.449 | 0.07 | <0.001 | |
| | Equation (2) | $a_1$ | 1.508 | 0.34 | <0.001 | 3.15 |
| | | $a_2$ | −1.046 | 0.01 | <0.001 | |
| | | $a_3$ | 1.045 | 9.34 | <0.001 | |
| | Equation (8) | $b_1$ | 0.059 | 0.002 | <0.001 | 2.83 |
| | | $b_2$ | −5.094 | 0.48 | <0.001 | |
| | | $b_3$ | 19.809 | 0.32 | <0.001 | |

Figure 3 presents the bias and RMSE by age class. Estimation with the GADA method showed a distribution of bias around zero for both parameters with no consistent trend.

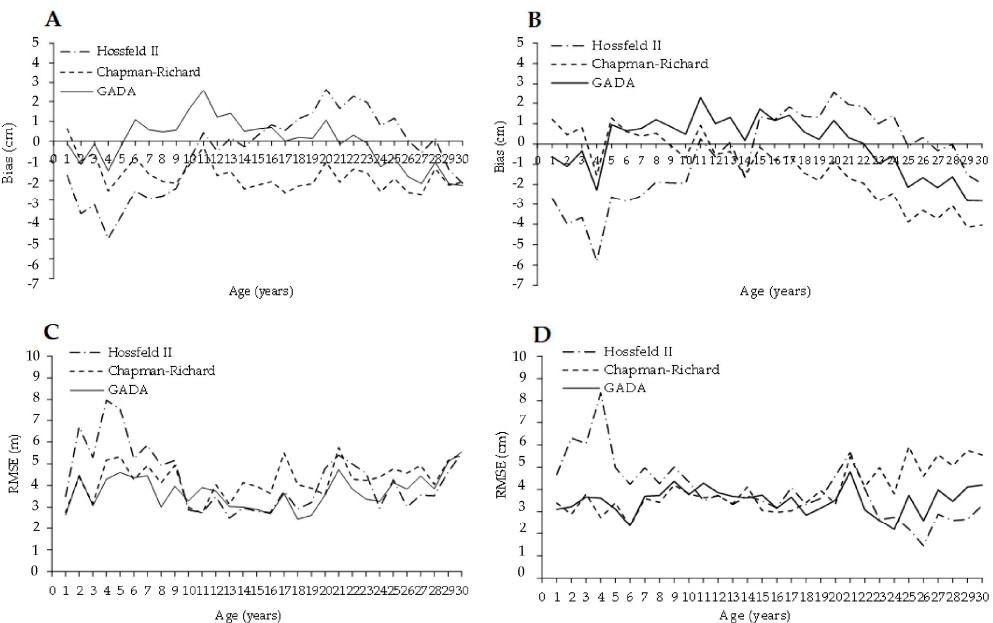

**Figure 3.** Bias (**A**,**B**) and root mean square error (RMSE; (**C**,**D**)) by age class for H (**A**,**C**) and DBH growth (**B**,**D**) estimated with the Equations (2), (3) and (8) (Chapman-Richards, Hossfeld II, and GADA formulation, respectively).

The GADA method displayed a lower bias in all age classes, especially at young ages. The lowest RMSE values were obtained with the GADA model for both *H* and *DBH* growth

for all age classes. RMSE values were higher for all ages when the SI was calculated with Equations (2) and (3). The trajectory of these two parameters corroborates that the best model was the polymorphic model generated by the GADA method (Figure 3). With the GADA procedure based on Chapman-Richard's model, both parameters $a_1$ and $a_3$ are dependent on the on-site quality, and the error structure is included in the interaction procedure, being:

$$\text{Height } H_1 = H_o \left[ \frac{1 - exp^{-0.1088*t_1}}{1 - exp^{-0.1088*t_0}} \right]^{(-12.5939+42.0932/X)} \tag{21}$$

$$\text{Diameter } DBH_1 = DBH_o \left[ \frac{1 - exp^{-0.0589*t_1}}{1 - exp^{-0.0589*t_0}} \right]^{(-5.0938+19.8095/X)} \tag{22}$$

where

$H_1$ = predicted height (m) at age $t_1$ (years),

$H_0$, $t_0$ = initial dominant height and age.

Height

$$X = \frac{1}{2} \left\{ lnH_0 - 12.5939 * L_0 \pm \sqrt{[lnH_0 - 12.5939 * L_0]^2 - 168.3729 * L_0} \right\}$$
$$L_0 = ln\left[1 - exp^{-5.0938*t_o}\right] \tag{23}$$

Diameter

$$X = \frac{1}{2} \left\{ lnH_0 - 5.0938 * L_0 \pm \sqrt{[lnH_0 - 5.0938 * L_0]^2 - 79.2383 * L_0} \right\}$$
$$L_0 = ln\left[1 - exp^{-5.0938*t_o}\right] \tag{24}$$

However, the model's bias shows the tendency of potential underestimation in older age classes (>23 years) when using GADA functions (for both dominant *H* and *DBH* growth), hence resulting in a potentially low asymptote. With this consideration, Laurel SI was constructed and divided into five classes. Class one (I) refers to the best site quality, and class V to the poorest quality for *H* and *DBH* growth. The GADA model was used to fit SI curves for H (from 12 m to 20 m). The site classes have two-meter steps and *DBH* growth (from 10 to 22 cm) at a reference age of 10 years (Figure 4).

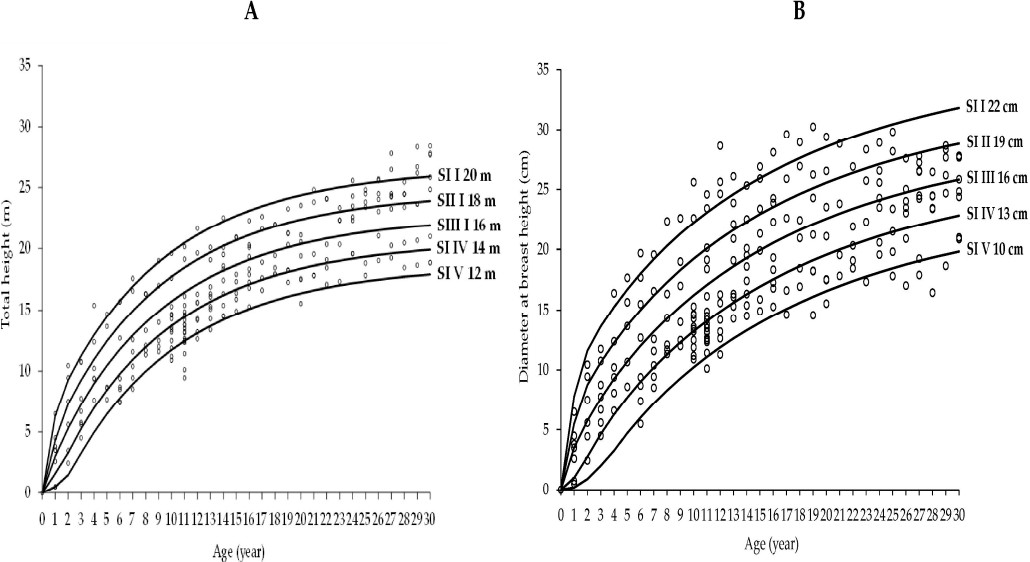

**Figure 4.** Site index curves by site class (I to V) for Laurel in the Ecuadorian Amazon upper basin ((**A**) *H*: total height) and (**B**): *DBH* growth, using the GADA model (reference age was 10 years).

### 3.3. Estimation of Laurel Total Volume and Yield

The tested models for estimating Laurel's total volume are ordered according to their AIC value (Table 4).

**Table 4.** Goodness of fit statistics of models predicting stem volume of Laurel in agroforestry systems in the Amazonian upper basin, Ecuador.

| Model | MSE | $R^2_{Adj}$ | Parameter | Estimator | SE | AIC |
|-------|-----|--------|-----------|-----------|-----|-----|
| Spurr potential | 0.12 | 0.96 | a | 0.00007 | <0.00 | −11.66 |
| | | | b | 1.44231 | 0.05 | |
| Schumacher-Hall (allometric) | 0.05 | 0.94 | a | 0.00036 | <0.00 | −7.28 |
| | | | b | −0.08518 | 0.09 | |
| | | | c | 2.48693 | 0.08 | |
| Australian formula | 0.05 | 0.94 | a | −0.17367 | 0.02 | −7.21 |
| | | | b | −0.00068 | <0.00 | |
| | | | c | 0.02476 | <0.00 | |
| | | | d | 0.00004 | <0.00 | |
| Incomplete generalized combined variable | 0.09 | 0.88 | a | −0.30835 | 0.02 | −6.60 |
| | | | b | 0.03523 | <0.00 | |
| | | | c | 0.00001 | <0.00 | |
| Honer | 0.17 | 0.87 | a | −343.79 | <0.00 | −6.02 |
| | | | b | 29876.01 | 0.03 | |
| Spurr with an independent term | 0.19 | 0.87 | a | 0.00795 | 0.01 | −5.9 |
| | | | b | 0.00004 | <0.00 | |
| Newnham | 0.78 | 0.49 | a | 0.24198 | 0.02 | −4.48 |
| | | | b | 0.000001 | <0.00 | |
| | | | c | −0.01904 | 1.23 | |
| | | | d | −1995.99 | 822.10 | |
| Spurr | 0.19 | 0.43 | a | 0.00045 | <0.00 | −0.67 |

MSE = mean squared error; $R^2_{Adj}$ = adjusted determination coefficient; SE = standard error; AIC = Akaike information criterion.

Based on this criterion, the best equation was the Spurr potential model (Equation (12)), which showed the best fit in terms of $R^2_{Adj}$ = 0.95 and the lowest AIC value. The Laurel total volume can be expressed as:

$$v = 0.00007 * \left( DBH * H \right)^{1.44231} \tag{25}$$

where *v* is Laurel tree volume in m³ for over-bark DBH of 5 cm or more, and H is the total height (m). The MAI and PAI were estimated using the Spurr potential model to calculate Laurel stem volume, as shown in Figure 5. BORA increased from 14 (25.30 cm of DBH) to 24 years (18.12 cm of DBH) with the decrease of SI in the Amazon upper basin.

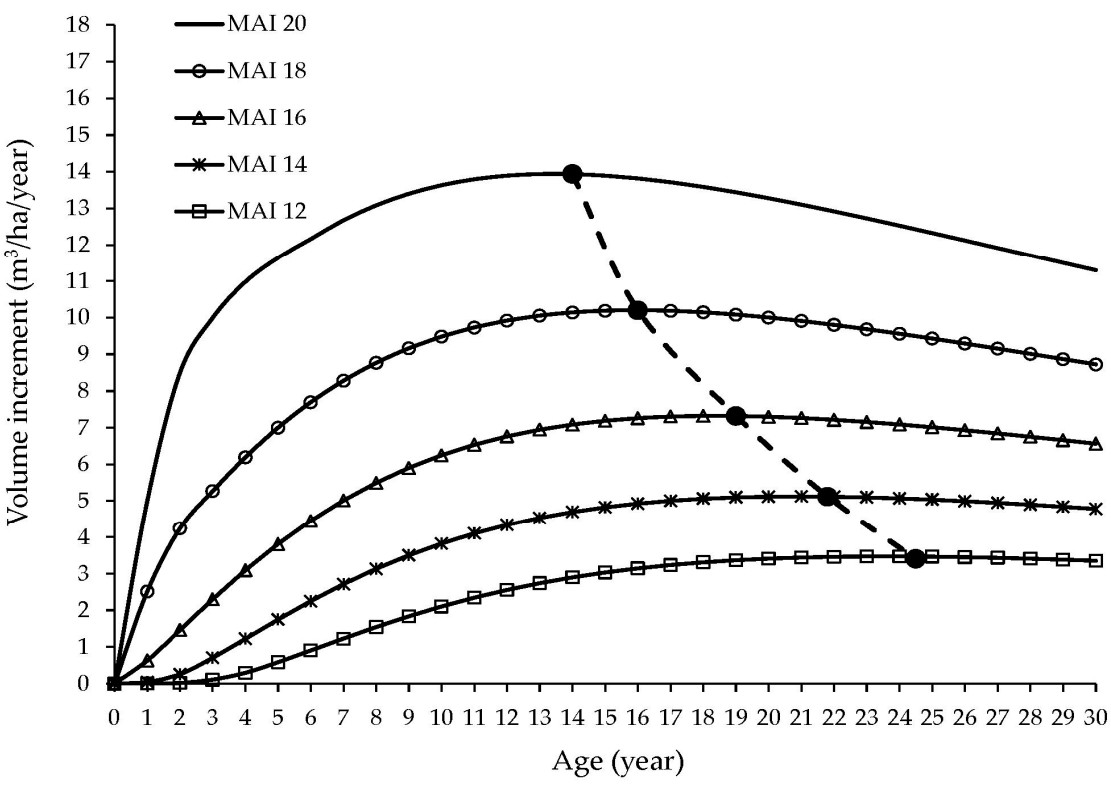

**Figure 5.** MAI over time and the intercept with the PAI (black circles) for volume increase at different ages and SI at ten years. The dashed line represents the development of biological optimal rotation age with a density of 300 Laurel trees ha$^{-1}$ at the Amazonian upper basin.

## 4. Discussion

### 4.1. Competition Index in Laurel Agroforestry System

The densities recorded in the agroforestry systems studied ranged from 80 to 398.4 trees per ha. According to Pretzsch [53], a stand density index SDI = 250–500 ha$^{-1}$ is considered a low tree population density for fast-growing, monospecific trees, resulting in little or no competition between trees and non-significant relationships between diameter growth and competition index (*RS*) (Figure 2). Pretzsch [53] highlighted that the exponential reduction of growth with increasing competition and stand density is considered to be a very simple relationship. Gonzalez de Andrés et al. [54] emphasized that the facilitation and reduction of competition depend on environmental conditions. Thus, the site indices shown by Laurel in the Amazon are lower than the average SI shown on the lower coast of Ecuador. Thus, facilitation and intraspecific competition would be lower in medium-quality SI as shown in the high Amazon [53]. Therefore, the *RS* was not considered to develop the diameter growth model and the total volume for Laurel (see below).

### 4.2. Height and Diameter Laurel Growth

For Laurel SI expressed as height classes, Alder and Montenegro [14] and Somarriba et al. [22] highlighted the need for shortening the measurement intervals in the sampling plots from four years to two or three years, especially due to the high growth rate of this species at youth stages. This issue was considered in the present investigation. However, relatively high values of bias and RSME at the age of 4 years are still recorded using the GADA method. Alder and Montenegro [14] found that the increase in diameter and the total height of Laurel is greater during the early life stages (less than 4 years of age) in the West Ecuadorian lowland coast region. The same pattern of Laurel tree growth was found in the study region (<5 years). Chapman-Richard's equations for *H* had a much greater bias in almost all age classes, especially at younger ages, and RMSE was also considerably larger for all age classes. Therefore, the best adjustment for the *H* model was found with

the GADA procedure by generating polymorphic curves, as compared to the Chapman-Richard's and Hossfeld II equations. Nevertheless, the projected potential asymptote might be too low for trees at any age, as trees older than 23 years are underrepresented (Figures 3 and 4A). Including more trees older than 23 years would be recommended to improve the SI model. However, agroforestry systems in the research zone are generally harvested at ages 14 to 24 years old. Moreover, the GADA method's advantages in the current study are comparable to those reported in agroforestry systems by Gea-Izquierdo et al. [55] and Cañadas-López et al. [38], among others, which allows for improving the description and simulation of Laurel biological growth. The variation of Laurel SI could be explained by the soil conditions prevailing in the Amazon region. Thus, Grau et al. [56] found that nitrogen availability can limit forest size and biomass in the Amazon region. In nutrient-poor sites, dominant trees often reduce their height growth [57–59].

On the other hand, the relationship of diameter increments and *RS* (%) (Figure 2) showed that a low tree density of the agroforestry systems could coexist mainly for a better distribution of resources and that stand growth is indirectly influenced by the chemical and physical characteristics of the soils [56]. With these H and DBH implications in the Amazon region, several studies have shown large differences in height, diameter growth, and yield for Laurel according to site quality in Ecuador and other countries (Table 5).

Table 5. Summarizes a review of growth in height, diameter, and yields of Laurel at a given age in different countries.

| Author | Country | Age (Years) | Best Site Height (m) | Best Site DBH (cm) | Worst Site Height (m) | Worst Site DBH (cm) | Best Site Volume (m$^3$ ha$^{-1}$) | Worst Site Volume (m$^3$ ha$^{-1}$) |
|---|---|---|---|---|---|---|---|---|
| Current study | Ecuador Amazon Region | 10 | 19.7 | 22.2 | 11.7 | 10.2 | 136 (300 trees ha$^{-1}$) | 21 (300 trees ha$^{-1}$) |
| | | 14 | 22.2 | 25.3 | 14.2 | 13.3 | 195 (300 trees ha$^{-1}$) | 41 (300 trees ha$^{-1}$) |
| | | 24 | 25.2 | 30.1 | 17.2 | 18.1 | 300 (300 trees ha$^{-1}$) | 83 (300 trees ha$^{-1}$) |
| Cañadas-López [30] | Ecuador Amazon region | 10 | | | 14.6 | | | |
| Alder and Montenegro [14] | Ecuador coast lowland region | 10 | 28.0 | | 18.2 | | | |
| | | 14 | | | | | 296 (300 trees ha$^{-1}$) | 83 (300 trees ha$^{-1}$) |
| Guamán [60] | Ecuador Amazon region Zamora Chinchipe | 31 | | | | 32.3 | | |
| Bílek et al. [61] | Ecuador Río Pitzara coast lowland region | 1.2 | 5.4 | | | | | |
| Mora [62] | Ecuador Sto. Domingo Coast lowland region | 6 | | | | | | 70 (708 trees ha$^{-1}$) |
| Briceño et al. [63] | Tropical Dry Forest, Tolima Colombia | 64 | | | | 205.5 | | |
| Martínez et al. [64] | Colômbia La Espirrilla | 7 | | | | | | 40 (357 trees ha$^{-1}$) |
| Giraldo et al. [65] | Colombia in higher altitudes | 50 | 30 | | 16 | | | |
| Sebbenn et al. [66] | Brazil São Paulo | 23 | 8.8 | 11.7 | | | | 14.1 |
| Greaves and McCarter [67] | Central and South America | 20–30 | 35 | 55 | 30 | 40 | | |
| Somarriba et al. [22] | Central America | 9 | | | | | | 48 (204 trees ha$^{-1}$) |
| Parresol and Devall [21] | Panamá | 14 | | | 15.7 | | | |

**Table 5.** *Cont.*

| Author | Country | Age (Years) | Best Site | | Worst Site | | Best Site | Worst Site |
| | | | Height (m) | DBH (cm) | Height (m) | DBH (cm) | Volume (m³ ha⁻¹) | Volume (m³ ha⁻¹) |
|---|---|---|---|---|---|---|---|---|
| Somarriba et al. [68] | Costa Rica TalamancaPanamá Changuinola | 5 | 21 | 28.5 | 15 | 15.6 | | |
| | | 5 | | | | | 114 (300 trees ha⁻¹) | 74 (300 trees ha⁻¹) |
| Somarriba and Beer [69] | Costa Rica Province Limón | 34 | | | | | 690 (68–290 trees ha⁻¹) | 298 (68–290 trees ha⁻¹) |
| Lamprecht [70] | Costa Rica Turrialba | 10 | 12.5 | 16 | | | | |
| Heuveldop et al. [71] | Costa Rica Turrialba | 8 | 14.0 | 26.7 | | | | 14.6 |
| Johnson and Morales [72] | Costa Rica Turrialba | 13 | 19.6 | 22.2 | | | | |
| | Costa Rica Turrialba | 3 | 5.2 | 8.9 | | | | |
| Hummel [73] | Costa Rica low elevation Atlantic coast | 10 | 17.1 | 21.3 | | | | |
| Beer et al. [74] | Costa Rica Atlantic coast | 10.5 | 15 | 24 | | | | 77.6 (278 trees ha⁻¹) |
| Johnson and Morales [72] | Costa Rica Atlantic coast | 24 | 29.3 | 37.8 | | | | |
| Hallé, et al. [75] | Tropical rain forest | 5 | 12 | 10.5 | | | | |

This fact contradicts Somarriba and Beer [69], who stated that Laurel showed very little variability in growth under the large range of variabilities of ecological conditions in Costa Rica. In Ecuador, the height, diameter growth, and total volume production in the coastal region were higher than that reported for the upper Amazonian region. The differences could be explained by the prevailing condition in the Amazon region. Cañadas [3] pointed out that clay-rich soils, which are susceptible to compaction, site altitude, the high aluminum content of soils, and high rainfall, can adversely affect the growth of this species. Giraldo et al. [65] found that the more acidic the soil is, the less the Laurel grows. Bergmann et al. [76] established that Laurel is a nutrient-demanding species. Later, Somarriba et al. [68] described a great variation in Laurel growth based on soil conditions in Costa Rica. Thus, plots located on the lowlands had a higher SI than on the hills. However, some lowland plots might show a low SI despite good fertility.

*4.3. Laurel Total Volume*

In relation to the Laurel total volume, the agroforestry systems are characterized by low tree density, and the dominant crown concept was not applicable because tree spacing reduces competition, generating a single layer of dominant and co-dominant trees [55]. According to Cañadas-López et al. [38], under these conditions, stem shape is most strongly influenced by the low density of trees than by the dominant height. In a study conducted in Costa Rica on Laurel volume, Somarriba and Beer [69] did not provide statistical dimension data on 146 trees, so we did not have a range vision of the sampled trees used for the volume model. According to Kleinn [77], if, for example, a volume model has been developed for diameters between 10 and 50 cm, it would not be a good idea to use this model for trees larger than 50 cm or less than 10 cm of DBH. This could generate tremendous errors and be useless in describing this parameter of Laurel for the Central America subcontinent, as Somarriba et al. [22] intended.

Under Amazonian nutrient-poor soil conditions, according to Grau et al. [56], tree growth is less limited by light on sites where N and P are restrictive compared to nutrient-rich soils (Ecuadorian coast region). For this reason, the competition factor was not considered for estimating volume models. With this regard, the Spurr potential model (Equation (12)) was the best-fitted model to determine the total volume of Laurel trees within traditional agroforestry systems at the Amazon upper basin. Tree volume gave reliable results for the determination of total volume. This model is suitable for Laurel trees with the dimensions exposed in Table 1. However, the Spurr potential model generated should be tested with Laurel trees from other regions to validate this model across regions. Alder and Montenegro [14] developed a Laurel total stand volume of a 10 cm top diameter over bark and volume under bark calculated to a minimum top diameter of 20 cm related against Lorey's height for the west lower coastal region of Ecuador. Even though the Laurel volume equation using the Spurr potential model is useful to estimate individual tree volume, it would overestimate total stand volume compared to Alder and Montenegro's equation since the first does not adjust individual tree volume by the effect of tree competition in agroforestry systems.

To determine Laurel standing volume, Cañadas-López [78] found that the Pain function is a very good approximation to describe it for Laurel agroforestry systems in the Protected Forest Sumaco. Similarly, Cañadas-López et al. [79] conducted an empirical study of Laurel in agroforestry systems, where it was shown that with data on six points (diameter measurement) well distributed along the tree stem, the spline function interpolation generated good results (compared with Kleinn [77]). Laurel stem biomass accumulation in either reforested or forest from natural regeneration management is considered an important tool to better understand the role of such local systems, the Amazon rainforest, and global climate change [80].

*4.4. Laurel Yield in Agroforestry System*

Under the best local conditions in the research Amazon region, a MAI of 14 m$^3$ ha$^{-1}$ year$^{-1}$ was obtained at 14 years (300 trees ha$^{-1}$) see Figure 4. Meanwhile, Cañadas-López et al. [57] reported a MAI of 13.92 m$^3$ ha$^{-1}$ at 9 years of age with a tree density of 418 trees ha$^{-1}$. These results agreed with Heuveldop et al. [71] in Costa Rica (15 m$^3$ ha$^{-1}$ year$^{-1}$) at 600 m elevation, with 278 trees ha$^{-1}$. Piotto et al. [81] reported a MAI of 16 m$^3$ ha$^{-1}$ year$^{-1}$ for Laurel in the Atlantic lowlands of Costa Rica. In contrast, Somarriba and Beer [69] observed a superior MAI (20 m$^3$ ha$^{-1}$ year$^{-1}$) in the canton Turrialba, province of Limón, Costa Rica. Beer et al. [74] established a Laurel MAI of 9 m$^3$ ha$^{-1}$ year$^{-1}$ at 10 years of age in Costa Rica.

The lowest growth rate (MAI: 3.5 m$^3$ ha$^{-1}$ year$^{-1}$ and yield of 83 m$^3$ ha$^{-1}$ at 24 years) was found at site elevations between 900 and 1160 m, conditions that limit Laurel growth [3,30]. At these elevations by the foothills of volcano Sumaco, the altitudinal gradient allows the presence of cold mountain forests at the lowest elevations. Therefore, differences in forest structure and biomass stocks may be found among sites over short distances [82,83]. Under this consideration, Laurel production was markedly lower than that reported by Heuveldop et al. [71], who registered a MAI of 15 m$^3$ ha$^{-1}$ year$^{-1}$ at age 7 years at 600 m.

Laurel BORA increases from 14 to 24 years with decreasing SI and increasing altitude above sea level in the upper Amazon basin. A similar Laurel BORA (20 years) was obtained by Salas and Valencia [84] in Colombia and by Hudson [85] in Vanuatu. Johnson and Morales [72] mention a BORA of 24 years in Costa Rica, while Greaves and McCarter [67] report 20 to 25 years in Costa Rica, and Vega [86] reported 25 years in Suriname. In the lowland coast of Ecuador, Alder and Montenegro [14] established a BORA ranging from 5 years on high SI sites up to more than 30 years on the worst sites, while on average sites, a rotation cycle of 11 to 16 years can be obtained. The above data are below those reported by Beer [87,88] and Somarriba and Beer [69], who established a Laurel BORA of 34 years in agroforestry systems in Costa Rica. The wood yield obtained from Laurel trees and the establishment of a natural tree life cycle in the upper Amazon basin will be the main inputs for this approach's cost and benefit evaluation.

The Environmental Ministry of Ecuador (EME) [89], according to Normative 125, established a Laurel minimum cutting diameter (MCD) in the Amazon region and foothills of 30 cm DBH. Considering the Laurel BORA from 14 years to 24 years mentioned above for different SI sites, trees would reach only 25.30 cm DBH at the best SI sites and 18.12 cm DBH at the worst sites at the end of the recommended BORA. Trees would require 30 years and over 40 years at these sites, respectively, to reach the 30 cm MCD established by law. It is necessary to emphasize that in Ecuador, there is equipment for processing volumes of less than 20 cm DBH for veneer production [14]. Thus, the Laurel MCD imposed in the Upper Amazon region is well above the BORA of the species in this region. Nevertheless, Cañadas [3] mentioned that Laurel is considered a secondary forest species and a rather ephemeral tree. It is possible that EME [89] Laurel MDC in the research zone refers to trees that are now slipping in an over-mature stage, and they may yield a reduced stem volume as losses due to defects or decay might offset growth [74] since older trees have a higher chance of being exposed to the rust fungus (*Puccinia cordiae*), especially under high humidity conditions, developing cankers on the trunk [67].

The combination of silvicultural modeling and simulation tools is basic to understanding the growth and production of Ecuadorian tropical forests, which will allow for maximum wood production [90]. The Laurel BORA would make sense if the sale of timber would be conducted by timber cubing and not as it is performed on a per-tree basis [37,38,57]. As there is no cubing of the timber, the price at the producer level is set by diameter. The larger the DBH, the higher the price, and vice versa. Thus, a 30 DBH Laurel tree would cost 30 US dollars. According to Figure 4B, reaching those diameter dimensions would take 10 more years at the best site. However, smallholders do not wait long to harvest trees in the research zone, mainly due to economic needs and the decrease in the

yields of cash crops. This detail leaves the BORA without any justification. An alternative to applying the BORA would be linking smallholders Laurel producers with the industry for pulp production [6] or veneer production [14]; then, the BORA would make sense.

## 5. Conclusions

The alternative hypothesis put forward in this study was accepted due to the better fit of the SI, DBH growth curve, and volume model. The best model for Laurel's SI was found using the GADA method, especially due to a reasonable description of the *H* growth rate of this species at the juvenile stage. The GADA model asymptote seems to be low in trees older than 23 years. Therefore, modeling the SI for Laurel agroforestry systems using *H* is not always straightforward, especially when there are no Laurel trees of older classes. Because the densities of the agroforestry systems studied were low, the competition-sensitive DBH growth model could be used as a substitute for the *H*-SI to approach the true value of the SI. The Laurel BORA increases from 14 to 24 years with decreasing SI and increasing elevation in the upper Amazon basin. The Laurel MCD imposed by EME should be decreased from 30 cm based on the results presented in this research in relation to BORA in Agroforestry systems and would suggest a review of the Laurel MCD from the primary forest in the research zone at least. The wood yield obtained from Laurel trees and the establishment of a natural tree life cycle will be the main inputs for this approach's cost and benefit evaluation. This information contributes to comparing the options of sites planted by seedlings vs. natural regeneration based on the tree life cycle for tropical forest reforestation.

**Author Contributions:** Á.C.-L. carried out the field data compilation, Á.C.-L., P.G.-T., S.B.-G., C.V. and B.M.-T., analyzed, designed the tables and figures, and wrote the drafts Á.C.-L., P.G.-T., S.B.-G., C.V. and B.M.-T., reviewed drafts of the paper, and improved the statistical analysis. Á.C.-L., P.G.-T., J.d.J.V.-H. and C.W., reviewed drafts of the paper Á.C.-L., J.d.J.V.-H. and C.W. All authors have read and agreed to the published version of the manuscript.

**Funding:** This research received no external funding.

**Acknowledgments:** We thank the General Director of the National Institute of Agricultural Research (INIAP) for providing the necessary logistics to carry out this project and Jorge Elis, Specialist of the Natural Sciences Sector Program of UNESCO, for the collaboration provided under the United Nations Yasuní Program and the Federation of Indigenous Organisations of Napo (FOIN) on behalf of Bertila Avilés for the coordination with communities and data collection. We thank the Faculty of Biology of the Central University of Ecuador, Gorky Gómez Díaz for providing the funds for the publication.

**Conflicts of Interest:** The authors declare no conflict of interest.

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
