# Peer review of "Laurel Regeneration Management by Smallholders to Generate Agroforestry Systems in the Ecuadorian Amazon Upper Basin: Growth and Yield Models"

_forests, doi:10.3390/f14061174_

Round 1
Reviewer 1 Report
GENERAL COMMENTS:
- This study tries to model the growth and yield of local species. The tree growth and yield model itself is not a new scientific approach. However, this study can be an example that the model is still relevant today in the context of regeneration management. This article also raises a traditional natural regeneration and agroforestry system as a local forest management practice.
- Please consider adjusting according to the detailed comments below.
ABSTRACT
- Lines 23-25: In my understanding, you tested 3 equations, and GADA is one of them. Please improve the sentence.
- Please make sure that you have followed the guideline related to the keywords.
INTRODUCTION
- This section is quite concise and well-organized. However, it seems to jump in some parts when introducing various issues. For example, please consider adding linking sentences between paragraphs 2 and 3.
MATERIAL AND METHOD
- Section 2.1: Please mention the land status of the study site whether state or private forest.
- Lines 112-113: please add more sentences to briefly elaborate on “natural regeneration” and “traditional management”.
- Lines 114-118: What is the specific type of agroforestry system at the study sites? Agrisylvopasture (as mentioned in the 1st paragraph of the introduction)? If so, please indicate the pastoral system in the species composition.
- What consideration was taken in determining the plot sample location/distribution? Please add this information in section 2.2 (site index).
- If the Laurel trees grow in natural regeneration, how do you measure the age of the tree samples? Please add this information in section 2.2 (site index).
- Line 136: How do you consider the different tree ages of each plot when measuring the site indices?
- Line 194: written 2.2, should be 2.3?
- Line 195: What consideration was taken in determining the 195 trees sample over 226 plots? Please add this information in section 2.3 (volume estimation).
- Do you split the samples into the model fitting and model validation? If so please mention it.
RESULTS
- Figure 2: It shows a relatively high value of bias and RSME at the age of 4 years. Please make sure that you have discussed it in the discussion section.
- Figure 4: Those are nice curves. However, presenting 5 SI in one figure make the reader confuse to evaluate the curve of MAI dan PAI of each SI. I would suggest presenting the curve of each site index separately (would be 5 paired curves).
DISCUSSION
- Lines 385-375: do you mean maximum MAI (at the SI 20)? If so in my opinion it should be emphasized in this figure.
- How is the harvesting cycle applied by the forest owner? How different is it from the optimal biological cycle presented in Figure 4? I suggest including this issue in the discussion.
-
CONCLUSIONS
- Please make sure that you have answered the hypothesis mentioned in the introduction.
- Please consider shifting Line 416-419 to the end of conclusion section.
Author Response
Dear Reviewer 1
We thank you for your time taken to review the manuscript and agree that the comments certainly have helped us to improve our manuscript’s content.
Below we are providing answers to your questions.
Thank you in advance for your help.
Alvaro
Reviewer 2 Report
The manuscript entitled “Laurel regeneration management by smallholders to generate agroforestry systems in the Ecuadorian Amazon upper basin: growth and yield models” reports Laurel (Cordia alliodora Ruiz & Pav. Oken) site index and diameter growth models developed from measurement data from 226 sample plots in Ecuador, as well as selection of the best fitted stem volume model for further projections of yield and estimation of the optimal rotation age based on 195 sample trees. Considering importance of Laurel in local agroforestry systems, yet lack of relevant growth models for the species in the studied region, development of new models is necessary for accurate growth predictions. The study attempts to model dominant height and diameter growth testing 2 commonly used base functions and generalized algebraic difference approach (GADA) form of Chapman-Richards function. Authors tested eight stem volume models to describe total stem volume.
I find the topic relevant and important at the regional scale. New accurate growth functions help to more precisely evaluate site quality and stand development in the future. The studied material seems valuable and appropriate for such purpose. The modelling approach is rather well described, yet description of the study material seems somewhat insufficient. Namely, total number of measured height-age and diameter-age pairs are not presented, and selection of dominant trees should be more thoroughly described. Do ‘minimum of 40 tallest Laurel trees in each sample plot’ refer to 100 tallest trees per ha?
A confusing aspect of the manuscript is that authors claim to fit Site index curves for diameter and height (e.g. lines 87 – 88). To my knowledge, site index is usually expressed in terms of dominant height as a reasonably proxy for stand quality, yet diameter is not used for such purpose due to e.g. dependence of stand density. Therefore, I suggest carefully check throughout the text that Site index is related only to dominant height.
Also, I suggest to revise the hypotheses (lines 87 – 92), since none of both are further addressed in the results. There are no measures presented in the methodology how ‘no significant differences between the observed total volume and the predicted values of eight models studied’ would be detected (only model fit statistics are mentioned in lines 211 – 214 for choosing the best fit volume model). In my opinion, also the claimed objective of the study ‘to analyse the implications of the estimated productivity for Laurel from the models developed ’does not reflect what has really been presented in the results. Namely, the main objectives seem to include also the development of dominant height-age and diameter-age curves, as well as volume model for yield projections.
The results section relatively sufficiently describes the model fitting statistics and selection of the final fitted models. However, the discussion and conclusions should be revised and supplemented. The discussion section mainly compares the obtained growth curves and projected volume and yield with the corresponding figures in other studies, yet deeper reflection of the current results is missing. The authors stress importance of estimating biologically optimum rotation age (BORA) for Laurel in the agroforestry system, hence I suggest to elaborate more on this issue extending the paragraphs in lines 402 – 419.
2
Meanwhile, the conclusions lack straightforward key messages from the study. I believe that some conclusions and practical implications related to BORA and changes in minimum cutting diameter are key findings of this study, yet are not presented here. In addition, I find sentences in lines 427 – 433 redundant and not conclusions related to the current results. Such statements may be included in the discussion of introduction supported by appropriate references.
Overall, the manuscript is rather easy to read, yet language editing may be advised.
Considering the study material and seemingly correctly applied modelling approach, I suggest major revision to address the drawbacks and some inconsistencies listed below.
Some comments.
L30. Too superficial statement. Does the minimum cutting diameter should be lowered or raised? Provide specific information based on the obtained results.
L41. Please, be more specific and define what is meant with ‘contemporary realities’.
L123. Why exactly minimum of 50 Laurel trees per plot? As I wrote above, more detailed description of study material is missing (e.g. total number of measured trees, range of number of trees per sample plot etc).
L148. I suggest to add a reference for guide curve method.
L202. Not ‘Several volume estimation models’, but exact number of models. Please, be more specific.
L205. Please provide reference and version for Statistica program (similar to the one for R program in line 184).
L224 and Figure 4. Probably I have missed something, but what is a period length used for calculating PAI? Is it one year?
Figure 2. Please be consistent throughout the paper in model names – the authors compare Hossfeld II and Chapman-Richard functions, not Hossfeld and Richard-Chapman. Check the whole text for inconsistencies.
Figure 2 and Figure 3A. The model bias shows the tendency of potential underestimation in older age classes (> 23 years) when using GADA functions (for both dominant H and DBH), hence resulting in potentially too low asymptote. I suggest to address this in the results and discussion.
L268. I suggest to add that site classes have two-meter step.
Figure 3 caption. Site index curves are only in subplot A. As mentioned above, I don’t think that DBH curves should be called site index.
Lines 358 – 361. I think that this limitation of not including any kind of competition index should be addressed also for diameter-age growth curves. If the competition is
3
not considered, it should be clearly justified in the discussion or in the methodology, why it has not been done.
Lines 368 – 370. Please provide reference to the statement. Anyway, I don’t see any added value from including this sentence in the discussion.
Line 410. What is MAE?
Author Response
Dear Reviewer 2
We thank you for your time taken to review the manuscript and agree that the comments certainly have helped us to improve our manuscript’s content.
Below we are providing answers to your questions.
Thank you in advance for your help.
Alvaro

Reviewer 3 Report
The authors have selected an important topic "Laurel regeneration management by smallholders to generate 2 agroforestry systems in the Ecuadorian Amazon upper basin: 3 growth and yield models". To meet the scientific standards and the level of the journal (Forests) I would like to suggest
1: Kindly elaborate the study area
2: Kindly provide more clear map
3: References must be up to date.
4: If possible provide some pics from ground zero.
5: Kindly check your numbering of the sub paras in Materials and Methods section.
Author Response
Dear Reviewer 3
We thank you for your time taken to review the manuscript and agree that the comments certainly have helped us to improve our manuscript’s content.
Below we are providing answers to your questions.
Thank you in advance for your help.
Alvaro

Round 2
Reviewer 1 Report
Dear Authors,
Thank you for your revised manuscript. The manuscript seems been improved incorporating Reviewers; comment. However, I failed to evaluate the response to my comments in detail because you did not enclose your response file.
Therefore please submit your response file through this system or email to the Assistant Editor.
Thank you very much for your cooperation.
Author Response
Good morning Dear Reviewer 1
A thousand apologies for the error, I am sending you the response file.
Kind regards
Alvaro

Reviewer 2 Report
The authors have largely incorporated the previous suggestions in the revised version of the manuscript entitled ‘Laurel regeneration management by smallholders to generate agroforestry systems in the Ecuadorian Amazon upper basin: growth and yield models’. I find issues related to proposed research hypothesis and objectives to be sufficiently averted. The conclusions have been improved to reflect the practical value of the research. Nevertheless, the discussion still includes unclear paragraphs. Namely, authors have tried to justify lack of competition index in L335 – 338 according to my previous suggestions. However, I find the reasoning kind of awkward and illogical. Authors claim that light is not the limiting factor opposite to the N:P ratio, but in the next sentence they state that quadratic diameter was “inversely proportional to stand density”. The latter statement literally means that there is decreasing mean DBH with increasing stand density! Consequently, this is opposite justification to actually HAVE a competition index! In addition, the meaning of the sentence in L343 – 346 is unclear. Why diameter-age curve should be an alternative to Site index curve?
Nevertheless, I actually find rather straightforward justification in L366 – 367, why competition is not considered in the volume estimations. I think this may also apply to the DBH modelling! In addition, you show in the Methods, that the stand density in the sample plots is low (80 to 398.4 trees per ha). Refer to ir here.
Overall, the language of the manuscript has been improved, yet I suggest to consistently use past or present tense (it is very inconsistent now). Considering the drawbacks of the discussion and some smaller issues listed below, I suggest another major revision.
Some comments:
L23 – 25. I suggest to clarify that you use the basic form of Chapman-Richards and Hossfeld II models and the GADA form of the Chapman-Richards function.
L29 – 30. Please clarify that the Spurr model was the most appropriate for the volume predictions.
L31 – 32 Maybe it is possible to suggest better limit according the results (27 or28 cm?)
L60 – 62 I suggest rephrasing e.g. ‘Porro et al. [9] highlighted several promising agroforestry initiatives in the Amazon that have been insufficiently..’
L64. No comma after ‘that’.
L67. Correct to ‘concept of Site index (SI)’.
L104. I suggest rephrasing to ‘’The soils belong to a large group known as …’
L159. I think here should be ‘equation’s parameters’
L200. ‘three-mean’ or ‘tree-mean’?
L201 ‘DBH’ instead of ‘DBH-age’.
L222. You still should add version to the Statistica program if it is relevant.
L249. ‘resumes’ instead of ‘resume’.
Figure 2. In general, the plot axis and their titles seems to be too small. There is still ‘Richard-Chapman’ instead of ‘Chapman-Richard’ in the plot legends.
L268. ‘polymorphic’ instead of ‘polymorphs’?
L286 ‘too low asymptote’ instead of ‘low asymptote’.
Figure 3. The same as for Figure 2 – too small letters. + I believe there should be cm instead of m for two-step classes in the plot B.
L309. ‘24’ instead of ’24-’
L329 – 330. The asymptote is not low for trees older than 23 years. The projected potential asymptote might be too low for tree at any age. There may be some model underestimations for projections over the age of 23 years. Here the authors may, for instance, suggest to include more older trees in the fitting dataset to overcome this issue.
L451 – 454. Please clarify. From the stated it looks like it is reasonable to keep the trees growing longer, since the price is increasing with the DBH.
L454. ‘basis’ or ‘justification’?
Author Response
Dear Reviewer 2
We thank you for your time taken to review the manuscript and agree that the comments certainly have helped us to improve our manuscript’s content.
Below we are providing answers to your questions.
Thank you in advance for your help.
Sincerely
Alvaro
